# OpenReview forum: "Trust, But Attribute: Tracing Impact of Data on Trustworthiness in Supervised LLM Fine-Tuning"
_NeurIPS.cc/2025/Workshop/Reliable_ML — NeurIPS 2025 - Reliable ML Workshop_

### Official Review · Reviewer_THs3 · 2025-09-18
**Appropriate and strong paper that needs more comparisons**

**Rating:** 7
**Confidence:** 3

**Review:**

# Summary. What the paper claims, how it does it, and the main results.

This paper presents a novel method for quickly and precisely fine-tuning a language model to improve trustworthiness after the original SFT process. Specifically, for each trustworthiness aspect, the paper first uses determinantal point processes (DPP) to identify a small but diverse subset of SFT samples that contributed disproportionately to the model’s poor performance on that aspect (theoretically arguing in the process that the subset must be small to maintain model performance). It then uses a proximal Bregman response function (PBRF) objective to slightly increase the model’s loss on those particular harmful data points while leaving its performance and parameters as constant as possible. Experiments show that the method is fast and successfully increases trustworthiness with limited impact on primary performance (perplexity).

# Strengths. Novelty, rigor, empirical/theoretical quality, clarity, relevance to reliability with imperfect data.

* Judged on its own merits (in other words, aside from the needed comparisons mentioned below), this is overall a strong paper. While there were some ways that the presentation could be streamlined in the methodology section, the paper is well-written and benefits from fair mathematical precision. The methodological insights are illuminating, the evaluation is fairly thorough, and the presentation is fairly clear.
* The paper seems moderately novel, with the closest comparison I know of being “Correcting Large Language Model Behavior via Influence Function” (Zhang et al., 2024), discussed more below.
* The paper has clear relevance to the workshop’s theme.

# Weaknesses / Limitations. Missing comparisons/ablations, unclear assumptions, proof gaps, failure modes, scope limits.
* The greatest weakness of this paper is lack of comparison to any other methods, even including naive retraining: the paper runs full retraining and reports relative time complexity results but not relative performance (trustworthiness) results, instead just claiming based on prior work that such retraining often fails. The reader thus is left unsure whether naive retraining would outperform the proposed method from the perspective of final trustworthiness and performance, if not efficiency.
* Furthermore, the method of “Correcting Large Language Model Behavior via Influence Function” (Zhang et al., 2024) seems very closely related to this paper. It is cited in the related work section, but dismissed as solving a different problem from “actively repair[ing] or adjust[ing] model parameters to improve the trustworthiness of the model.” I do not see why: while Zhang et al. (2024) work with a slightly different formulation of trustworthiness (they do not work with log odds in particular), and while they don’t do DPP, they do use PBRF for what seems like a very similar goal of reducing harmful behavior probability.
* If there were very strong inherent justification for using this paper’s log-odds definition of trustworthiness, the missing comparison to Zhang et al. (2024) would be reasonable; however, I did not find any explanation for that setup. Why not just use arbitrary trustworthiness metrics that could be implemented as log-odds if the user so chose?

# Suggestions for Authors. Specific things that would improve the paper:

* Line 134: What does theta^post being the “optimal” parameters after SFT mean? Doesn’t SFT have to balance pre-training and post-training accuracy? Could this just be “fuzzified” to theta^post being the parameters after SFT, whether optimal or not?
* The nature of the trustworthiness metrics F_j was very unclear to me when I first read through the methodology section, as it took a long time to become clear to me that this was a log-odds-style metric. I believe that fact could be introduced right away rather than waiting until line 176 – we probably wouldn’t define F_j in terms of v if it’s not log-odds anyway, so I’m not sure why we should wait to introduce that fact (though I still think that the setting could use more abstract metrics than log-odds in the first place).
* Equation 2 is an approximation, not a definition (we would need new symbols other than F_j if it were a definition). To avoid miscommunication like this, true definitions should use defined-to-be-equal-to signs (equals sign with a triangle above it or a colon in front). This is fairly misleading.
* The methodology section does not make very clear that (m, o, p) triplet data is an assumed input to the method; for a while while reading, I was unsure whether the method would itself show a way to categorize data as o or p. Mentioning that the method “assumes access to” this data would help with clarity, and then explaining the reasonability of this assumption would prevent lingering doubts from the reader.
* There were a few grammar issues here and there – capitalizations, unnecessary commas, etc. I would recommend putting the paper through your chatbot of choice and asking it to identify and fix these.
* Mentioning the advantage of Bregman divergence over the much more widely known KL for this context, or why KL isn’t applicable, could help the reader.
* I understand that it is desirable to introduce gamma before talking about how to find subsets, but by the time I got to section 3.5, I had been wondering for a long time what the subsets were and how we found them, since they’re fundamental to the method as a whole. I think 3.5 could be before 3.4, in short.

# Ethics (if applicable). Note any concerns (about privacy, fairness, misuse, sensitive data use) and suggested mitigations.
N/A

---

### Official Review · Reviewer_6os3 · 2025-09-19
**Review of submission #48**

**Rating:** 7
**Confidence:** 3

**Review:**

The paper addresses the problem of degrading trustworthiness during Supervised Finetuning. The authors propose a two-stage, computationally efficient repair procedure. First, they trace failures in trustworthiness metrics back to specific fine-tuning samples using gradient similarities and then they select a diverse set of training samples via a deterimental point process.
Finally they perform an "unlearn"-like gradient ascent step on the selected harmful samples.

They experiment on Pythia and Qwen models and demonstrate big improvements in trustworthiness while retaining the model's perplexity.

* Strengths:
1) The proposed method is very efficient compared to re-training or re-finetuning the model on a dataset that excludes the examples.
2) The method exhibits strong empirical results that work across various models and sizes.

* Weaknesses
1) The method relies on the existence of labeled bad/untrustworthy examples

Overall it is an interesting result and I think it would be a nice addition to the workshop.